# Extracellular Hsp90 Binds to and Aligns Collagen-1 to Enhance Breast Cancer Cell Invasiveness

**DOI:** 10.3390/cancers15215237

**Published:** 2023-10-31

**Authors:** Pragya Singh, Varshini Ramanathan, Yang Zhang, Irene Georgakoudi, Daniel G. Jay

**Affiliations:** 1Department of Developmental, Molecular and Chemical Biology, Graduate School of Biomedical Sciences, Tufts University School of Medicine, Boston, MA 02111, USA; pragya.singh@tufts.edu (P.S.); irene.georgakoudi@tufts.edu (I.G.); 2Department of Biomedical Engineering, Tufts University School of Engineering, Medford, MA 02155, USA; varsh@mit.edu (V.R.); yang.zhang@tufts.edu (Y.Z.)

**Keywords:** cancer invasion, metastasis, extracellular matrix, Collagen-1, extracellular Hsp90, LOXL2, second harmonic generation imaging, Ganetespib, STA-12-7191

## Abstract

**Simple Summary:**

Breast cancer cells secrete Hsp90, a protein that, inside of cells, regulates the function of hundreds of proteins, but outside of cells, extracellular Hsp90 (eHsp90) can activate a subset of proteins that promote invasion, the first step of metastasis. Blocking eHsp90 in mouse models inhibits metastasis, and we sought to understand how this occurs. Prior studies have predominantly focused on eHsp90 in cancer invasion within the immediate vicinity of the primary tumor, specifically its role in invading outside the epithelial compartment. However, eHsp90’s role in cancer invasion across the extended connective tissue after the cells have crossed the boundary of the epithelial compartment remains unknown. We show here that eHsp90 directly binds to and aligns Collagen-1 fibers, a major structural component of connective tissues, which, when aligned, form highways that allow efficient cancer migration. Our study suggests that the Hsp90 dimer, in its open state, binds to Collagen-1 molecules to align the fibers, which results in enhanced breast cancer invasion through the Collagen-1 matrix. Knowing this could help us propose experiments to test eHsp90 inhibitors for therapeutically targeting metastatic breast cancer.

**Abstract:**

Cancer cell-secreted eHsp90 binds and activates proteins in the tumor microenvironment crucial in cancer invasion. Therefore, targeting eHsp90 could inhibit invasion, preventing metastasis—the leading cause of cancer-related mortality. Previous eHsp90 studies have solely focused on its role in cancer invasion through the 2D basement membrane (BM), a form of extracellular matrix (ECM) that lines the epithelial compartment. However, its role in cancer invasion through the 3D Interstitial Matrix (IM), an ECM beyond the BM, remains unexplored. Using a Collagen-1 binding assay and second harmonic generation (SHG) imaging, we demonstrate that eHsp90 directly binds and aligns Collagen-1 fibers, the primary component of IM. Furthermore, we show that eHsp90 enhances Collagen-1 invasion of breast cancer cells in the Transwell assay. Using Hsp90 conformation mutants and inhibitors, we established that the Hsp90 dimer binds to Collagen-1 via its N-domain. We also demonstrated that while Collagen-1 binding and alignment are not influenced by Hsp90’s ATPase activity attributed to the N-domain, its open conformation is crucial for increasing Collagen-1 alignment and promoting breast cancer cell invasion. These findings unveil a novel role for eHsp90 in invasion through the IM and offer valuable mechanistic insights into potential therapeutic approaches for inhibiting Hsp90 to suppress invasion and metastasis.

## 1. Introduction

Breast cancer is a devastating disease that kills approximately 43,000 people annually in the US [1]. While therapies have improved outcomes for breast cancer if treated while localized, breast cancer-caused mortality is most often attributed to metastasis [1,2]. Unfortunately, no targeted therapies are currently available to specifically target cancer’s metastatic spread, making the development of anti-metastasis drugs a critical and unmet medical need. Targeting metastasis is challenging because of its complex steps, which are not completely understood. While intervention at any step to prevent the metastatic spread would be beneficial, blocking the initial step of invasion before the tumor spreads beyond a single location holds promise as a valuable strategy. The work of our lab and that of others have suggested that extracellular Hsp90 (eHsp90) may serve as a drug target to reduce cancer invasion and metastasis [3,4,5,6,7,8,9,10,11,12,13,14,15,16,17,18,19].

Our lab’s work and that of others have shown that various cancers secrete Hsp90 into the extracellular environment (termed eHsp90) [20]. Hsp90 is primarily an intracellular ATPase chaperone protein vital for hundreds of intracellular proteins’ functions and cell homeostasis and is upregulated and dysfunctional in several cancers [20,21]. Eustace and colleagues presented the first evidence that cancer cells secrete Hsp90 and showed that eHsp90 increased the in vitro invasiveness of cancer cells in fibrosarcoma and breast cancer cells [3]. In this study, eHsp90 activated matrix metalloproteinase 2 (MMP-2), an extracellular protease known for its role in degrading extracellular matrix (ECM) proteins and invasiveness [3]. Since this discovery, eHsp90 has been shown to interact with multiple extracellular proteins. Interestingly, a large subset of Hsp90’s interacting proteins in the TME are ECM proteins (fibronectin) [23,24,25], or ECM-modifying proteins, such as MMP-2 [3,8,10,11,18,26], MMP-9 [18], Tissue Plasminogen Activator (tPA) [27], and LOXL2 [28]. Through these interactions and those with other membrane-bound proteins, eHsp90 has been shown to increase the invasiveness of several types of cancer cells, including breast (MB231, SUM159, MCF-7, SKBR3, MB453, MB435, MB361 MB468, BT474, and T47D) [3,11,19,27,28,29,30], fibrosarcoma (HT1080) [3], melanoma (B16) [9,17], colorectal (HCT-8) [29], prostate (PC3) [15,19], bladder (T24) [19], and glioblastoma (A172) [27].

eHsp90’s role in cancer cell invasion has been studied in vitro using Matrigel [3,5,9,10,13,19,31] which mimics the composition of the basement membrane (BM) [32,33]. The BM, primarily composed of Collagen-IV and elastin, is a form of ECM protein that resembles a sheet-like structure and outlines the epithelial cell compartment, separating it from interstitial connective tissue [32,33]. During invasion, cancer cells breach the 2D BM matrix to exit the epithelial compartment and encounter the 3D-ECM i.e., IM present in the connective tissue [34,35,36]. The IM is a structurally and compositionally different form of the ECM that resembles a mesh-like structure and is primarily composed of Collagen-1 interspersed with fibronectin and mesenchymal connective tissue cells, which must be traversed to reach the circulatory system and continue metastasizing [34,35,36].

eHsp90’s role in increasing invasion through the IM has not yet been established. The current study examined the role of eHsp90 in the invasion of breast cancer cells through the IM’s primary component, the Collagen-1 matrix, and investigated the mechanism by which this occurs [37,38,39]. An alteration in the Collagen-1 matrix from thin, loose, and unaligned fibers to aligned, straightened, and bundled fibers surrounding the tumor can occur, which serves as a significant indicator of progressively increasing tumor invasiveness in both mouse models and human breast cancer samples [37,38,39]. This alteration in collagen fibers is clinically defined as the tumor-associated collagen signature (TACS), which ranges from 1 to 3 [37,38]. Notably, a higher TACS number corresponds to progressively enhanced tumor invasion [37,38]. Particularly, TACS-3 is characterized by a change in the alignment of collagen fibers relative to other fibers and the tumor itself, demonstrating a highly invasive phenotype surrounded by aligned fibers [38,40]. The presence of aligned collagen signature alone, independent of other factors, was shown to be able to predict patient survival in breast cancer [40]. It is known that these aligned collagen fibers create a highway-like structure, enabling the efficient escape of migratory tumor cells from their primary location [38,39,41,42,43,44,45].

eHsp90’s role in increasing invasion through the IM has not yet been established. The current study examined the role of eHsp90 in the invasion of breast cancer cells through the IM’s primary component, the Collagen-1 matrix, and investigated the mechanism by which this occurs [37,38,39]. An alteration in the Collagen-1 matrix from thin, loose, and unaligned fibers to aligned, straightened, and bundled fibers surrounding the tumor can occur, which serves as a significant indicator of progressively increasing tumor invasiveness in both mouse models and human breast cancer samples [37,38,39]. This alteration in collagen fibers is clinically defined as the tumor-associated collagen signature (TACS), which ranges from 1 to 3 [37,38]. Notably, a higher TACS number corresponds to progressively enhanced tumor invasion [37,38]. Particularly, TACS-3 is characterized by a change in the alignment of collagen fibers relative to other fibers and the tumor itself, demonstrating a highly invasive phenotype surrounded by aligned fibers [38,40]. The presence of aligned collagen signature alone, independent of other factors, was shown to be able to predict patient survival in breast cancer [41]. It is known that these aligned collagen fibers create a highway-like structure, enabling the efficient escape of migratory tumor cells from their primary location [38,39,42,43,44,45,46].

In this study, we demonstrated that eHsp90 plays a crucial role in enhancing in vitro breast cancer invasiveness through the Collagen-1 matrix by directly binding and aligning Collagen-1 fibers within the matrix. Furthermore, we provide evidence that eHsp90 binds to Collagen-1 through its N-domain. The alignment of Collagen-1 fibers induced by eHsp90 occurs through the open conformation state of Hsp90 rather than the closed state, and this activity is independent of its ATPase function.

## 2. Materials and Methods

### 2.1. Materials

pcDNA-3 FLAG-Hsp90α wild-type (WT), pcDNA-3 FLAG-Hsp90α D93A, and pcDNA-3 FLAG-Hsp90α E47A were generous gifts from Professor Leonard Necker (National Cancer Institute, Bethesda, MD, USA). GST-Hsp90-N (9-236) (plasmid #22481), GST-Hsp90-M (272-617) (plasmid #22482), and GST-Hsp90-C (623-732) (plasmid #22483) were obtained from Addgene. LOXL2 (LOXL2-3923H) was obtained from Creative Biomart. proLOXL2 (39AO) was obtained from R&D Systems. Novobiocin was obtained from Sigma Aldrich, St. Louis, MO, USA (1476-53-5), and STA-12-7191 (7191) was obtained from (Synta Pharmaceuticals, West Conshohocken, PA, USA).

### 2.2. Cell Lines and Culture

The cell lines 293T (ATCC CRL3216) and MDA-MB231 (MB231) (ATCC HTB-26) were maintained in DMEM supplemented with 10% FBS and 1% penicillin/streptavidin (P/S) (referred to as complete DMEM). SUM159 cells were a kind gift from Dr. Charlotte Kuperwasser (Tufts University, Boston, MA, USA). SUM159 cells were maintained in F-12 media supplemented with 5% FBS, 5 μg/mL insulin, 0.5 μg/mL hydrocortisone, and 1% P/S (complete F12 media). All cells were grown at 37 °C with 5% CO_2_.

### 2.3. Condition Media (CM) and Concentrated Condition Media (CCM) Collection

MB231 cells (20,000/cm^2^) were counted and plated in complete DMEM. After 24 h, the cells were washed with PBS (3X), and the medium was changed to FluoroBrite DMEM (Thermo Fisher A1896702, Waltham, MA, USA) without FBS, called serum-starved (SS) media. Cells were maintained in SS media for 48 h at 37 °C and 5% CO_2_. The media were collected and subjected to a first spin at 1000 RPM for 3 min at 4 °C to collect floating cells and debris as pellets. Half of the supernatant c/a CM was stored at −80 °C. The rest of the supernatant was subjected to spinning in an Amicon Ultra-15 centrifugal concentrator (Millipore UFC9011024, Burlington, MA, USA) unit with a 10 KDa molecular weight cut-off (MWCO) at 4000 RPM for 20 min, resulting in a 10× concentration of the CM (CCM). The CCM was aliquoted and stored at −80 °C.

### 2.4. Collagen-1 Binding Assay

Prior to starting the Collagen-1 binding assay, 100 μg of protein was biotinylated using an Abcam biotinylation kit (Abcam Ab201795, Cambridge, UK) following the manufacturer’s protocol. Once biotinylated, excess unbound biotin was removed, and the biotinylated protein was concentrated to 0.5 mg/mL using Abcam’s antibody concentration and cleanup kit (Abcam Ab102778). For the Collagen-1 binding assay, the surface of the 96-well plate was coated using rat tail Collagen-1 diluted in 0.1% acetic acid to twice the desired concentration to be used and neutralized with an equal volume of neutralization buffer (10× PBS, 0.4 M NaOH,250 mM HEPES, dH_2_O) [46] to form a final Collagen-1 solution of 0.02 μg/μL with 1X PBS and 25 mM HEPES. Subsequently, 50 μL of Collagen-1 solution was added to each well. Collagen-1 polymerization was carried out for 2 h at 37 °C. The plates were dried in a vacuum chamber under sterile conditions overnight to obtain a coated surface. If not immediately used, the plates were stored at 4 °C and used within five days. The plates were washed three times with 1X PBS before the experiment. Next, the plates were blocked using a no-protein blocking buffer (Pierce 37572, Appleton, WI, USA) for 1 h at RT. After blocking, 50 μL of PBS containing proteins and inhibitors was added to each well of the plate, and the plates were incubated for 1 h at RT with gentle rocking. After an hour, the plate(s) were washed with ELISA wash buffer (PBS with 0.1% Tween pH 7.4) 7–8 times. The plate was then treated with streptavidin–HRP (1:150,000) in 50 μL ELISA wash buffer and incubated for 30 min at RT. The washes were repeated 7–8 times. Next, 50 μL of TMB substrate (Thermo Fisher TMB N301) was added, and the absorbance was immediately read at 607 nm. The absorbance values were averaged, and the raw average absorbance values were normalized to the 0 nM FLAG-Hsp90α (containing only PBS) absorbance values.

GraphPad PRISM (Version 9.5.1) was used to determine the K_d_ values using non-linear regression based on one-site total binding. One-way ANOVA was performed with Tukey’s test to correct for multiple comparisons, and paired *t*-tests were performed for experiments with only two conditions. The *p*-value was summarized and represented (GP method) as follows—*p* < 0.1234 as “NS”, <0.0332 as ”*”, <0.0021 as “**”, <0.0002 as “***”, and <0.0001 as “****”.

### 2.5. SHG Imaging of Collagen-1 Gels and Fiber Variance Analysis

Rat tail Collagen-1 (Corning 354236, Corning, NY, USA) was used to set up Collagen-1 gels according to the protocol described by Roy et al. in a 24-well glass-bottom plate (Celltreat #229125) [46]. Polymerization was carried out for 2 h at 37 °C. The treatment conditions were added in PBS alone or spiked with 10 μL of CM or CCM. The volume of PBS with the treatment condition was equal to that of the Collagen-1 gel mix. The gels were incubated for five days at 37 °C. SHG images were acquired on the fifth day using a Leica TCS SP8 confocal microscope (Wetzlar, Germany) equipped with an fs laser (Insight, Spectra-Physics). Samples were excited with 920 nm light using a 40×/1.1 NA water immersion objective, and 12-bit, 512 × 512-pixel images with a 67 × 67–145 × 145 μm^2^ field of view (FOV) were obtained. SHG images were recorded in epi-illumination mode through a 460 (±25) nm bandpass filter. Z-stacks were acquired at a step of 0.5 or 1 μm over a depth of 70 μm starting above 200 μm from the base of the gel. Two to three stacks per gel were acquired. In each experiment, each condition/treatment was performed on three gels. All SHG imaging experiments were repeated at least three times, except for the experiment conducted with CCM in Section 3.3.

The extent of collagen alignment was determined by calculating the three-dimensional directional variance (3D variance) of each SHG image stack. The 3D variance analysis was performed using MATLAB code (MathWorks, Natick, MA, USA) adapted from an approach to quantify the 3D fiber organization described previously [47,48]. The SHG image stacks were filtered for collagen signals using a two-level Otsu threshold. Orientation vectors were calculated for each voxel within a 3D window, whose dimensions varied between 5 × 5 × 3 and 9 × 9 × 3 pixels between experiments to ensure that the xy face of the window was 2–3 times the approximate fiber width. The lateral-to-axial resolution ratio was as close to 1 as possible in each condition. These conditions were shown to yield optimal accuracy in the recovery of the corresponding fiber orientation vectors [47,48]. The window sizes varied due to variations in the size of the field of view selected for image acquisition. The 3D variance in the vector directions was then evaluated for each voxel in this 3D window using a 3-pixel radius disk kernel filter that was averaged over all voxels. The 3D variance was obtained as a metric of fiber alignment and always varied between 0 and 1, with lower 3D variance values corresponding to increased fiber organization and alignment.

To account for inter-experiment variation in collagen alignment and the parameters used in the image analysis due to lot-to-lot Collagen-1 and imaging condition variability, the fiber variance of each stack within the experiment was normalized to a single stack variance value of their respective negative control (no treatment—NT). That is, all variance values within an experiment were divided by the variance value of a randomly assigned stack of the NT treatment. The normalized variance data of separate experimental repeats were combined per condition and averaged. The statistical analysis was then performed for the combined experimental repeats to determine significant differences in conditions across experiments. The *p*-value was summarized and represented (GP method) as follows—*p* < 0.1234 as “NS”, <0.0332 as” *”, <0.0021 as “**”, <0.0002 as “***”, and <0.0001 as “****”.

### 2.6. Transwell Invasion Assay

MB231 and SUM159 cells (20,000/cm^2^) were plated in complete DMEM and complete F12 media, respectively, for 48 h at 37 °C and 5% CO_2_. After incubation, the cells were serum-starved in FluoroBrite DMEM in the case of MB231 cells and no-phenol DMEM/F12 in the case of SUM159 cells for 2 h at 37 °C and 5% CO_2_. Meanwhile, the Collagen-1 matrix in the Transwell chamber was rehydrated by adding SS media to both chambers for 2 h. After serum starvation, 50,000 cells and treatment conditions in 400 μL serum-free media were prepared with condition treatments and added to the top well of each Transwell. Then, 750 μL of chemoattractant media—DMEM containing 2% FBS (for MB231) and F12 media containing 1% FBS (SUM159)—was added to the bottom well after removing the hydration media. The cells were then allowed to invade through the Collagen-1 matrix for 8 h in the case of MB231 cells and 4 h in the case of SUM159 cells. At the end of incubation, cells that invaded the other side of the membrane were visualized by adding Calcein (Corning 354217) at 4 μM in PBS to the bottom well. The images of the invaded cells were captured by microscope at 485 nm. The captured images spanned a whole Transwell vertical section at 10× magnification. One image per well was created by overlaying approximately 10–12 images. Each condition was tested in at least 3 wells per experiment, and each experiment was repeated at least 3 times.

The cells that crossed through the Collagen-1 matrix and porous membrane, i.e., invading cells, were counted by ImageJ software (Version 2.9.0/1.53t) First, the background was subtracted at 50 pixels; then, the threshold for counting the particles (Yen) was set at 20 and 255 and converted to binary. Lastly, “analyze particles” was run to determine the count of invasive cells. One-way ANOVA was performed to determine the significance of the difference in the cell invasion count with Tukey’s correction, in which the mean of each condition was compared with all other means. The *p*-value was summarized and represented (GP method) as follows—*p* < 0.1234 as “NS”, <0.0332 as ”*”, <0.0021 as “**”, <0.0002 as “***”, and <0.0001 as “****”.

### 2.7. Mammalian Cell Transfection, Protein Expression, Lysis, and FLAG-Tagged Purification

First, 293T cells were plated in complete DMEM and incubated overnight at 37 °C and 5% CO_2_. Six hours prior to transfection, the medium was replaced with 25 μM chloroquine diphosphate. Approximately 3 μg of DNA/well in a 6-well plate was mixed with 2 M CaCl_2_ solution and then mixed with 2× HBS solution in a 1:1 ratio. The DNA and transfection reagents were allowed to form a complex for about 15 min at RT and added dropwise to the cells. Cells with transfection reagent were incubated for about 16–18 h at 37 °C. The media were then changed to fresh complete DMEM, and the cells were incubated for 48 h at 37 °C. After that, the cells were lysed with cell lytic buffer (FLAG M Purification Kit, Sigma Aldrich CELLMM2) containing an added 1X HALT protease inhibitor. The lysed cells were then spun down at 16,000 g for 15 min, and the supernatant was collected (cell lysate) and stored at −80 °C. Protein was purified following the protocol provided with the FLAG M Purification Kit. The FLAG M beads were incubated with the cell lysate for 5 h. After incubation, the supernatant was drained, and the beads were extensively washed with 10 mL 1× wash buffer 4 times. Next, the FLAG-tagged proteins were eluted using 2 mL 1× elution buffer in 5 batches. The purified proteins were quantified using silver staining and Western blotting.

### 2.8. Bacterial Protein Induction, Cell Lysis, and GST-Tagged Protein Purification

BL21(DE3) cells containing the bacterial vector with GST-tagged Hsp90 N, M, and C domains were grown until the OD600 value reached 0.6 in LB media containing 100 μg/mL ampicillin. The protein expression was then induced by adding 1 mM IPTG (isopropyl β-D-1- thiogalactopyranoside) at 30 °C for 5 h. The cells were collected for protein expression verification. The rest of the cells were pelleted by spinning at 5000 rpm for 10 min at 4 °C. The supernatant was discarded, and the pellet was saved at −80 °C until lysed. The pellet was lysed on ice using lysis buffer (which contained lysozyme, DNase A, RNase 1, and protease inhibitor) for 30 min, followed by sonication to break the DNA in the cell lysate. The bacterial lysate was centrifuged at 20,000 g at 4 °C for 30 min. The debris was discarded. Cell lysate supernatant was used to affinity purify GST-tagged protein following an optimized protocol based on the manufacturer’s instructions (Bio-Rad GST-tag purification kit 6200214, Hercules, CA, USA). The glutathione beads were incubated with the cell lysate supernatant for 5 h at 4 °C while spinning. The beads were extensively washed 6 times using 5 mL—1× wash buffer that was prechilled at 4 °C. Next, the elution was performed using 1× elution buffer (containing reduced glutathione) in a batch elution with 2 mL 1× elution buffer 5 times. The rest of the specifications were exactly as per the protocol provided by the manufacturer. The Western blotting and silver staining (Invitrogen SilverQuest Silver Staining Kit, Waltham, MA, USA) were performed to determine purity and concentration quantification.

### 2.9. Cell Viability Assay

A quantity of 100 μL of media with 10,000 cells was plated per well of a 96-well plate. The plate was incubated at 37 °C for the duration of the invasion assay. After incubation, the Pierce Cell Titer-Glo kit was used as instructed by the manufacturer. The luminescence was read, and raw values were plotted. One-way ANOVA with Tukey’s test was used to determine the significance of the difference in cell viability.

## 3. Results

This study stems from a prior study conducted in our lab that demonstrated the binding of eHsp90 to an inactive form of Lysyl Oxidase-2 (proLOXL2) present in the condition media of breast cancer cells [29]. An activated proLOXL2, i.e., LOXL2, is known to bundle and align collagen fibers by crosslinking them through oxidation of the amine residue of lysine present in the triple-helical structure of collagen [49]. In several cancers, LOXL2 is not only frequently upregulated, but its upregulation is also implicated in tumor invasiveness and metastasis [50,51,52,53,54]. Given that aligned Collagen-1 fibers enhance cancer cell invasion [38,55], we sought to investigate the role of eHsp90 in breast cancer invasion through the Collagen-1 matrix via its interaction with proLOXL2. Specifically, we aimed to examine whether eHsp90 could align Collagen-1 fibers by interacting with the inactive form of proLOXL2 and facilitating its conversion into the active LOXL2 form.

### 3.1. eHsp90 Increases Breast Cancer Invasiveness through Collagen-1

To establish the role of eHsp90 in breast cancer invasion through Collagen-1 (the major ECM component of the IM), we performed an invasion assay with MB231 cells, an aggressive, poorly differentiated, and invasive triple-negative breast cancer (TNBC) cell line. We employed a Collagen-1-coated Transwell setup to create a matrix barrier that cells crossed to invade physiologically (Schematic—Figure 1A). We also expressed and purified FLAG-Hsp90α WT in 293T cells to test the invasiveness caused by extracellular Hsp90 (Appendix A). We observed a concentration-dependent increase in MB231 invasion with the exogenous application of purified FLAG-Hsp90α WT protein in comparison with non-treated (NT) MB231 cells (Figure 1B). This observation indicates a role for eHsp90 in the invasion of MB231 cells through Collagen-1.

Conversely, we also explored the impact of 7191, a biotinylated form of Ganetespib, specifically targeting the extracellular population of Hsp90 due to its limited ability to penetrate cell membranes efficiently [28]. We chose 10 and 100 nM concentrations of 7191 because of its known binding affinity for Hsp90 (K_d_—8 nM), as well as our previous study investigating the effect of eHsp90 inhibition using 7191 to study cancer cell migration (including MB231 cells) through Matrigel without affecting cell viability [28]. The 7191 was dissolved in DMSO; therefore, the effect of 7191 on MB231 invasiveness was compared against MB231 cells treated with DMSO alone. We observed a dose-dependent decrease in MB231 cell invasion through the Collagen-1 matrix upon treatment with 7191. These observations supported the involvement of eHsp90 in promoting the invasive behavior of MB231 cells through the Collagen-1 matrix, while prior studies were conducted with Matrigel (which mimics the BM). In the current experiments (Figure 1C), we observed that 7191 decreased the invasiveness of MB231 below the baseline invasiveness of DMSO-treated MB231 in a concentration-dependent manner, confirming the significance of eHsp90 in promoting the invasiveness of MB231 cells through Collagen-1.

To establish the generalizability of eHsp90’s effects on invasion through Collagen-1, we extended our analysis to include a second TNBC invasive breast cancer cell line, SUM159. Upon treating SUM159 cells with FLAG-Hsp90α WT, specifically at a concentration of 5 nM, we observed an increase in Collagen-1 invasion compared with non-treated (NT) SUM159 cells. Furthermore, when exogenously added Hsp90 was inhibited using 7191 at a concentration of 10 nM, the increased invasion phenotype caused by eHsp90 was rescued, indicating that the increased invasiveness is a specific phenotypic change induced by eHsp90 (Figure 1D). Interestingly, when SUM159 cells were treated with 7191 at a 10-fold higher concentration, i.e., 100 nM, it led to a decrease in their invasion below the baseline invasiveness of NT SUM159 cells. This observation suggests that the decrease in invasion below the NT cell’s threshold may be attributed to the inhibition of basal eHsp90 secreted by SUM159 cells [28].

To ensure that cell proliferation or cell viability did not influence the observed changes in cell invasion, we conducted cell viability assessments by utilizing the CellTiter-Glo luminescence assay for both the MB231 and SUM159 cell lines (Appendix A). The CellTiter-Glo luminescence assay showed no difference in cell viability between different conditions in either cell line when the cells were incubated for the duration of the invasion assay. This observation showed that the observed variations in cell invasion numbers accurately reflected the extent of cell invasion through the Collagen-1 matrix rather than changes in cell proliferation or death.

Taken together, these observations show that eHsp90 promotes the invasiveness of breast cancer cells through the Collagen-1 matrix without significantly affecting cell viability or proliferation.

### 3.2. eHsp90 Does Not Increase Breast Cancer Invasion through Collagen-1 via proLOXL2 Activation

Understanding how eHsp90 promotes Collagen-1-based invasion would be valuable as we consider this pathway for therapeutic intervention. After the initial discovery of Hsp90’s secretion by cancer cells and pro-invasive interactors, we sought to identify additional extracellular proteins that interact with eHsp90. McCready et al. utilized mass spectrometry analysis of conditioned media of MB231 cells and identified several proteins, including proLOXL2, the inactive form of the Lysyl Oxidase-Like 2 proteins (LOXL2) [28].

The LOX family of proteins comprises secreted proteins that play a crucial role in the crosslinking of various ECM proteins [56]. Among them, LOXL2 has been shown to be upregulated and contribute to the invasion and metastasis of several cancers, including breast cancer [57,58,59]. One of the key roles of secreted LOXL2 is its ability to crosslink collagens, including the major component Collagen-1, leading to the bundling and increased alignment of Collagen-1 fibers [54,60]. Notably, this process requires the activation of LOXL2 from its inactive form, proLOXL2 (100 KDa), to its active form, LOXL2 (65 KDa) [60,61]. On the basis of these observations, it is plausible to hypothesize that eHsp90 binds to and activates proLOXL2, thereby facilitating the alignment of Collagen-1 fibers and ultimately promoting the invasion of MB231 through the Collagen-1 matrix.

To determine whether the increase in the invasion of breast cancer cells is a result of the alignment of Collagen-1 fibers induced by eHsp90 via proLOXL2 activation, we employed SHG imaging of Collagen-1 gels (Appendix A). SHG microscopy permits the non-destructive visualization of anisotropic structures, such as Collagen-1 fibers, in tissue samples, including breast cancer ones, without exogenous processing [38,62,63]. SHG has also been successfully utilized to visualize Collagen-1 fibers in gels [47,48]. We utilized SHG to visualize Collagen-1 fibers in gels and analyzed the 3D directional variance of fibers. Three-dimensional directional variance is a quantitative metric of the 3D organization or alignment of fibers, with increasing values from 0 to 1 representing collagen fiber organization changing from perfectly aligned to entirely disorganized fibers, respectively [47,48].

In our experimental setup, rat tail Collagen-1 gels were treated with purified proteins—proLOXL2 (containing the 100 KDa form only) or a combination of Hsp90–proLOXL2. The purified protein treatment was spiked with 10× CCM to ensure that Hsp90 and proLOXL2 had access to the extracellular factors that they might require to align Collagen-1. A set of Collagen-1 gels was also treated with only PBS and only CCM. The only PBS condition is considered NT in SHG imaging experiments, and only CCM was added to determine the baseline change in the alignment of Collagen-1 fibers because of CCM spiking. We observed that the CCM-only treatment did not alter Collagen-1 fiber alignment, nor did the CCM spiked proLOXL2 treatment result in a change in Collagen-1 fiber alignment. Surprisingly, quantitative comparison of fiber alignment of Collagen-1 gels treated with proLOXL2 with Hsp90 also did not reveal a significant difference in alignment in comparison with NT or proLOXL2 alone. These observations suggest that either Hsp90 cannot activate proLOXL2 to its active LOXL2 form or that Hsp90-activated LOXL2 in the presence of CCM is unable to align Collagen-1 fibers under these conditions.

To determine whether Hsp90 is capable of activating proLOXL2 (100 KDa form) to the active form LOXL2 (65 KDa), we incubated Hsp90 and LOXL2 (comprising both forms proLOXL2-100 KDa and LOXL2-65 KDa) together at 37 °C for 1 h and subsequently performed SDS-PAGE followed by silver staining of the gel (Appendix A). Surprisingly, our observations revealed that even at a concentration ten times higher than that of LOXL2, Hsp90 did not activate the proLOXL2 band to the LOXL2 band. These observations strongly suggest that Hsp90 does not activate proLOXL2.

We verified that Hsp90 increased Collagen-1 invasion of breast cancer cells by activating proLOXL2 via the Collagen-1 Transwell invasion assay (Appendix A). The MB231 cells in the upper chamber of the Transwell were treated with purified FLAG-Hsp90α WT, proLOXL2, LOXL2, and a combination of FLAG-Hsp90α WT and proLOXL2. After 8 h of incubation, the MB231 cells that invaded the Collagen-1 matrix were visualized using Calcein imaging. Quantification of the invaded cells revealed that treatment with Hsp90 and LOXL2 significantly increased the invasion of MB231 cells through Collagen-1 compared with NT MB231 cells. As expected, treatment with proLOXL2 alone did not induce a significant increase in MB231 invasion compared with the NT condition. Interestingly, the combination treatment of Hsp90 and proLOXL2 did result in a higher invasion of MB231 cells compared with NT MB231 cells. However, the difference between the invasiveness of MB231 cells treated with Hsp90 alone and the combination treatment of Hsp90 with proLOXL2 was insignificant. This result suggests that Hsp90 increases Collagen-1 invasion by breast cancer cells independent of proLOXL2 activation. Collectively, these experiments (Appendix A) provide clear evidence that Hsp90 does not activate proLOXL2 and Hsp90-proLOXL2 interaction, and subsequent activation to LOXL2 does not change the alignment of Collagen-1 fibers or the invasiveness of breast cancer cells through Collagen-1 fibers.

### 3.3. Hsp90 with Extracellular Factors Can Align Collagen-1

In the previously discussed experiment (Appendix A), in which Collagen-1 gels were imaged using SHG after being treated with purified proteins along with CCM, we tested one added condition in which Collagen-1 gel was treated with purified Hsp90 along with CCM (Figure 2A,B). Surprisingly, adding FLAG-Hsp90α WT treatment resulted in a significant increase in Collagen-1 fiber alignment compared with NT Collagen-1 gel. This is a novel finding since eHsp90 is conventionally not known to align Collagen-1 fibers in the TME, and how this may occur is not known. However, it is possible that another factor in the CCM carried out the alignment of Collagen-1 fibers here and was indirectly impacted by Hsp90’s presence in the treatment condition.

To verify Hsp90 as a primary driver of the Collagen-1 alignment observed in the previous experiment, we added varying FLAG-Hsp90α WT concentrations along with CM instead of 10× CCM to the polymerized Collagen-1 gels and performed SHG imaging and 3D directional variance analysis (Figure 2C,D). Collagen-1 gels treated with BSA (11 nM) were used as a negative control. We observed that the CM treatment of the Collagen-1 gel alone did not result in Collagen-1 alignment compared with the NT Collagen-1 gel. However, with an increasing concentration of Hsp90 along with CM, we observed an increase in the Collagen-1 fiber alignment starting at concentrations as low as 0.02 nM compared with NT. We also observed saturation in Collagen-1 fiber alignment at the 0.02 nM FLAG-Hsp90α WT concentration. The saturation of Collagen-1 fiber alignment at such a low concentration point to Hsp90’s efficacy at Collagen-1 fiber alignment in the presence of extracellular factors secreted by breast cancer cells in the CM. This result validates the novel finding that Hsp90, in conjunction with other factors secreted by breast cancer cells, aligns the Collagen-1 fibers efficiently.

### 3.4. Hsp90 Directly Binds to and Is Sufficient to Align Collagen-1 Fibers

Next, we investigated whether Hsp90 alone is sufficient to cause Collagen-1 fiber alignment in the absence of extracellular CM factors using SHG imaging of Collagen-1 gels. To examine this, we treated the Collagen-1 gels with varying concentrations of FLAG-Hsp90α WT in PBS and compared the fiber alignment in Collagen-1 gels to that of PBS alone (NT). We also treated Collagen-1 gels with BSA (negative control) and LOXL2 (positive control) mixed in PBS (Figure 3A,B).

We found that increasing concentrations of Hsp90 treatment resulted in a gradual increase in Collagen-1 fiber alignment compared with the NT Collagen-1 gels. However, a significant increase in Collagen-1 fiber alignment was observed only at and above a concentration of 2 nM of FLAG-Hsp90α WT. Notably, in the presence of CM factors, FLAG-Hsp90α WT induced an increase in Collagen-1 alignment that was observed at a 100-fold lower concentration of 0.02 nM. This suggests that while Hsp90 alone is sufficient to align Collagen-1 fibers, its efficacy is enhanced in the presence of breast cancer cell CM factors.

Conversely, we tested the effect of Hsp90 inhibition on Collagen-1 alignment using 7191 (Figure 3C,D). We did not observe a significant difference in Collagen-1 fiber alignment in Collagen-1 gels treated with FlAG-Hsp90α WT (20 nM) alone compared with gels treated with FlAG-Hsp90α WT (20 nM) with 7191 at a concentration of 10 nM. However, the assessment of Collagen-1 fibers revealed a significant decrease in the fiber alignment when gels were treated with a 100 nM concentration of 7191 with FLAG-Hsp90α WT (20 nM) in comparison with FLAG-Hsp90α WT (20 nM) alone. This observation suggests that the inhibition of Hsp90-induced Collagen-1 alignment by 7191 is concentration-dependent, further highlighting the specificity of Hsp90’s activity in Collagen-1 alignment.

To further confirm the specificity of Hsp90’s activity in aligning Collagen-1, we investigated the direct binding of Hsp90 to Collagen-1. We added biotinylated FLAG—Hsp90α WT protein to Collagen-1-coated wells and utilized biotin–streptavidin (conjugated with HRP) binding to measure the binding of Hsp90 to Collagen-1 (Figure 3E). We observed a direct correlation between the concentration of biotinylated FLAG-Hsp90α WT and its binding to Collagen-1, with a low dissociation constant (K_d_) value of 1.9 nM, indicating specific and tight binding (Figure 3F). Interestingly, the K_d_ value of 1.9 nM for Hsp90 binding to Collagen-1 corresponds well to the Hsp90 concentration of 2.2 nM, at which Collagen-1 alignment significantly increased compared with the NT treatment. Furthermore, we examined the impact of inhibiting Hsp90 using 7191 on Collagen-1 binding (Figure 3G). We observed that the combined treatment of 7191 and FLAG-Hsp90α WT at each concentration resulted in lower absorbance values compared with the corresponding concentration of the FLAG-Hsp90α WT-only treatment. These findings demonstrate the specificity and high affinity of Hsp90 binding to Collagen-1 and its effectiveness in aligning Collagen-1 fibers.

### 3.5. The ATPase Activity of Hsp90 Is Not Crucial for Collagen-1 Binding, but Inhibition of ATPase Activity Increases Hsp90’s Collagen-1 Fiber Alignment

We next wanted to understand the mechanism that Hsp90 employs to bind and align Collagen-1 fibers that ultimately promote breast cancer invasion through the Collagen-1 matrix. The 7191 used to study the effect of Hsp90 inhibition on Collagen-1 binding and alignment is a biotinylated form of pan Hsp90 inhibitor Ganetespib, which binds to Hsp90 at the N-terminal ATPase domain, effectively inhibiting its ATPase activity [25,28,64]. Since we observed that 7191 could inhibit Hsp90 binding to Collagen-1 and aligning Collagen-1 fibers and ultimately breast cancer invasion through Collagen-1, it is possible that Hsp90’s ATPase cycle and activity are necessary for Collagen-1 binding and alignment.

The ATPase cycle of Hsp90 begins with unbound Hsp90 in an open V-shape conformation. Once ATP and other ATP-like molecules occupy the ATP binding pocket in the N-terminal domain, Hsp90 undergoes a series of intermediate structural alterations before finally assuming an N-terminal closed conformation. The closed conformation remains until ATP hydrolysis is complete, releasing ADP and inorganic phosphate. Then, the N-terminal domain reopens, and Hsp90 resumes the V-shape open conformation [19,65,66,67,68].

To ascertain the importance of Hsp90’s ATPase catalytic activity in binding and aligning Collagen-1, we employed ATPγS, a slow hydrolyzing analog of ATP. By keeping Hsp90 in a closed conformation for an extended period, ATPγS effectively slows down the ATPase cycle, thereby inhibiting functions that rely on this cycle [68]. This approach allowed us to investigate the impact of inhibiting Hsp90’s ATPase activity on Collagen-1 binding and fiber alignment (Figure 4).

To investigate the significance of Hsp90’s ATPase activity in Collagen-1 binding, we conducted a Collagen-1 binding assay using FLAG-Hsp90α WT alone and FLAG-Hsp90α WT combined with ATPγS (100 μM). Our results, shown in Figure 4A, revealed no significant difference in the binding capacity of FLAG-Hsp90α WT to Collagen-1 with or without ATPγS. This indicates that Hsp90’s ability to bind Collagen-1 remains unaffected by its ATP hydrolysis capability, showing that ATPase activity is not critical for Hsp90–Collagen-1 binding.

Next, to determine whether Hsp90’s ATPase activity plays a significant role in Collagen-1 fiber alignment, we utilized SHG microscopy to examine Collagen-1 gels. Specifically, we compared the alignment of Collagen-1 fibers induced by FLAG-Hsp90α WT alone or in the presence of 100 μM ATPγS (Figure 4B,C). Consistent with the effect of ATPγS on Hsp90 binding to Collagen-1, we did not observe a significant difference in the Collagen-1 fiber alignment in gels treated with FLAG-Hsp90α WT at 2 nM premixed with ATPγS (100 μM) compared with FLAG-Hsp90α WT (2 nM) treatment alone. However, at a 10-fold higher concentration of 20 nM FLAG-Hsp90α WT, the combination treatment with ATPγS (100 μM) led to even more alignment of Collagen-1 fibers than 20 nM FLAG-Hsp90α WT alone.

These findings show that inhibiting Hsp90’s ATPase activity does not impede its ability to align Collagen-1 fibers. In fact, the increased alignment observed with ATPγS at a 20 nM Hsp90 concentration indicates that reducing Hsp90’s ATPase activity can enhance the efficiency of Collagen-1 fiber alignment. These results combined show that Hsp90’s ATPase activity is not crucial for Hsp90 binding or aligning Collagen-1.

### 3.6. Open Conformation Mutant Aligns Collagen-1 and Increases Breast Cancer Cell Invasiveness

While the conformational states of Hsp90 inside cells are known to facilitate its interaction with specific clients and co-chaperones, the role of conformational states of eHsp90’s interaction with extracellular clients and chaperones is less well understood [69,70,71]. How eHsp90’s conformational states influence Collagen-1 fiber binding and alignment is unknown. Moreover, transient conformations resulting from the Hsp90’s ATPase cycle did not show a difference in Hsp90-led Collagen-1 binding or fiber alignment; we wanted to see if fixing Hsp90 in a particular conformation would show a difference in Collagen-1 binding and aligning Collagen-1 and, subsequently, invasion. Therefore, we assessed whether a particular fixed conformation impacted Hsp90 binding and aligning Collagen-1 fibers.

We used Hsp90 proteins with point mutations in the N-domain that stabilized it in either an open conformation (FLAG-Hsp90α D93A) or a closed conformation (FLAG-Hsp90α E47A) [69]. The open-conformation mutant FLAG-Hsp90α D93A remained in the open state due to its inability to bind ATP or ATP analogs. By contrast, the closed conformation mutant FLAG-Hsp90α E47A was locked in a closed state, as it lacked the hydrolysis ability once ATP or ATP analogs were bound to it. We successfully purified FLAG-tagged Hsp90α D93A and Hsp90α E47A by expressing the recombinant protein in 293T cells and performing FLAG-tagged affinity purification from the cell lysate (Appendix A).

First, to assess the Collagen-1 binding ability of the fixed conformation Hsp90α mutants compared with the FLAG-Hsp90α WT, we performed a Collagen-1 binding assay using biotinylated recombinant proteins (Figure 5B). To induce the respective conformations in the mutant Hsp90α-D93A and E47A, the proteins were pre-incubated with 100 μM ATPγS. The Hsp90α WT, D93A, and E47A proteins mixed with ATPγS were also tested alongside Hsp90α WT and mutant proteins alone. We observed that all three recombinant proteins, including the WT, D93A, and E47A constructs, exhibited comparable Collagen-1 binding abilities. Furthermore, no significant difference in Collagen-1 binding was observed between Hsp90α-WT and the mutant Hsp90α D93A or E47A when the conformation was induced by adding ATPγS. This observation shows that the conformation of Hsp90 is not a factor determining its binding to Collgen-1.

Next, we investigated the effectiveness of the mutationally fixed Hsp90 conformation in increasing the alignment of Collagen-1 fibers. The FLAG-Hsp90α WT, D93A, and E47A proteins were preincubated with 100 μM ATPγS to induce the respective conformations. Subsequently, Collagen-1 gels were treated with the FLAG-tagged Hsp90α D93A and E47A mutants mixed with ATPγS, and their alignment was compared to Collagen-1 gels treated with NT, LOXL2 (positive control), and BSA (negative control) (Figure 5C,D). Consistent with our previous findings, Collagen-1 gels treated with FLAG-Hsp90α WT exhibited higher fiber alignment compared with the NT control. Interestingly, we noted that Collagen-1 gels treated with the open conformation mutant FLAG-Hsp90α D93A (with ATPγS) showed a significant increase in Collagen-1 alignment compared with the NT Collagen-1 gels. This increase in fiber alignment was comparable to the enhanced alignment observed in Collagen-1 gels treated with FLAG-Hsp90α WT. On the other hand, Collagen-1 gels treated with the closed conformation Hsp90 mutant FLAG-Hsp90α E47A (with ATPγS) did not exhibit increased Collagen-1 fiber alignment compared with the NT Collagen-1 gels. This outcome demonstrates that only the open conformation of Hsp90 is capable of aligning Collagen-1 fibers. By combining our previous observations regarding Collagen-1 alignment with the ATPγS experiment and the current experiment involving Hsp90 conformation mutants, we can conclude that while slowing down the ATPase cycle through ATPγS treatment does not diminish Hsp90’s ability to align Collagen-1 fibers, restricting Hsp90 to a closed conformation hinders its capacity to align Collagen-1 fibers. Consequently, we deduce that Hsp90 aligns Collagen-1 in its open conformation rather than its closed conformation.

Next, we used a Transwell invasion assay to determine whether the open vs. closed conformation of Hsp90 could increase the invasion of breast cancer cells through the Collagen-1 matrix (Figure 5E,F). MB231 cells were treated with purified FLAG-tagged Hsp90α WT, D93A, and E47A proteins at a concentration of 5 nM and were allowed to invade through the Collagen-1 matrix (Figure 5E). Following the incubation period, we observed an increase in the invasiveness of MB231 cells when treated with FLAG-Hsp90α WT or FLAG-Hsp90α D93A (open conformation) compared to the NT MB231 cells. On the other hand, MB231 cells treated with FLAG-Hsp90α E47A exhibited a level of invasiveness similar to that of the NT cells. Using the same experiment with SUM159 cells, we observed a similar pattern in SUM159 cell invasion through Collagen-1. Like MB231 cells, SUM159 cells treated with either FLAG-Hsp90α WT or FLAG-Hsp90α D93A exhibited enhanced invasion through the Collagen-1 matrix in comparison with NT SUM159 cells (Figure 5F). By contrast, SUM159 cells treated with FLAG-Hsp90α E47A did not demonstrate an increase in the invasion compared with the NT SUM159 cells. These results demonstrate that the open conformation of Hsp90 is crucial for promoting breast cancer cell invasion through the Collagen-1 matrix.

Overall, on the basis of the combined observation of experiments performed with mutationally restricted Hsp90α mutants, we concluded that while both conformations of Hsp90 are capable of binding to Collagen-1, it is exclusively the open conformation of Hsp90 that aligns Collagen-1 fibers, leading to a subsequent increase in the invasiveness of breast cancer cells through the Collagen-1 matrix.

### 3.7. GST-N-Domain Hsp90 Alone Binds to Collagen; However, Isolated N-Domain Is Insufficient in Increasing Invasiveness through Collagen-1

Given that Hsp90’s conformational changes are predominantly observed within its N-domain due to the ATPase cycle [67,72], we aimed to investigate the specific role of the N-domain in Collagen-1 binding. Additionally, we sought to investigate whether the N-domain is the domain that binds to Collagen-1 and contributes to Hsp90’s function of ultimately promoting the invasion of breast cancer cells through the Collagen-1 matrix.

To determine whether Hsp90’s N-domain is responsible for its direct interaction with Collagen-1 or whether other domains (M-domain and C-domain) bind to Collagen-1, we conducted a Collagen-1 binding assay with purified isolated Hsp90 domain proteins. The Hsp90 domain recombinant proteins were first expressed in BL21(DE3) cells and purified using GST affinity tag purification (Appendix A).

The purified GST-Hsp90 domain proteins, namely the N-domain, M-domain, and C-domain, were subjected to a Collagen-1 binding assay after biotinylation. We assessed the Collagen-1 binding capacity of individual Hsp90 domains against full-length FLAG-Hsp90α WT (positive control) and GST protein alone (negative control) (Figure 6B). FL- FLAG-Hsp90α WT exhibited efficient and concentration-dependent binding to Collagen-1, confirming its strong affinity, as observed in previous experiments. Conversely, GST protein alone displayed minimal Collagen-1 binding, as did the GST-Hsp90-M and GST-Hsp90-C domain proteins. Although the GST-Hsp90-M and GST-Hsp90-C domain proteins exhibited some level of concentration-dependent binding, their efficiency was significantly lower than that of the full-length FLAG-Hsp90α WT. This finding was unexpected, as the—domain of Hsp90 is well-known for its role in client interactions [21]. The GST-tagged Hsp90 N-domain exhibited a pronounced and concentration-dependent binding pattern to Collagen-1, mirroring the binding pattern observed with the FLAG-Hsp90α WT protein. On the basis of this observation, we concluded that Hsp90 primarily interacts with Collagen-1 through its N-domain.

We also utilized GST-tagged Hsp90 domain proteins to study whether a particular domain protein could enhance breast cancer cell invasion through Collagen-1 (Figure 6C). Here, the MB231 cells in the top well of the Transwell were treated with a 5 nM concentration of purified GST-tagged Hsp90 N, M, and C proteins. On the basis of SHG visualization and the quantification of Collagen-1 alignment experiment results, an additional condition tested for breast cancer invasion through Collagen-1 was purified GST protein only. This condition permitted us to segregate the effect on invasiveness due to Hsp90 domain proteins and GST protein and to use GST-induced invasion as a baseline for all three domains. The invasiveness caused by these proteins was compared to that imparted due to FLAG-Hsp90α WT as a positive control for invasion; 7191-treated wells were used as the negative control, along with NT MB231 cells. While an increase in the MB231 cells’ invasiveness through the Collagen-1 matrix was observed in cells treated with GST protein only compared with the invasiveness of NT MB231 cells, the difference in invasion between these two treatments was statistically insignificant. Similarly, the Collagen-1 invasiveness of MB231 cells treated with purified GST-tagged Hsp90 domain (N, M, and C) proteins displayed an insignificant difference compared with GST-tagged protein only. Even though it was statistically insignificant, the increase in the invasiveness that we observed with GST-tagged Hsp90 domain proteins compared with NT MB231 cells can be attributed to the GST tag itself. These observations suggest that none of the Hsp90 domain proteins can increase the invasiveness of MB231 cells through the Collagen-1 matrix.

Together, these results show that while the Hsp90 N-domain is sufficient to bind to Collagen-1, none of the domains of Hsp90 separately are sufficient to increase the invasiveness of MB231 cells through Collagen-1.

### 3.8. Hsp90 Dimerization Is Required for Collagen-1 Binding and Fiber Alignment

Hsp90, in its physiological state, exists as a homodimer, and the inhibition of dimerization disrupts its normal functions [73,74,75,76]. It has been established that the C-domain of Hsp90 is responsible for dimerization [76]. Given that Hsp90 adopts open and closed conformations as a result of its dimeric structure, we aimed to investigate whether dimerization is crucial for Hsp90 to bind to and align Collagen-1 or whether Hsp90 monomers alone can perform these functions. To assess the impact of dimer disruption on Hsp90-mediated binding to Collagen-1 and fiber alignment, we utilized novobiocin, an Hsp90 inhibitor that binds to the C-terminal hydrophobic pocket and prevents the dimerization of Hsp90 monomers [77,78].

In the Collagen-1 binding assay, we treated Collagen-1-coated plates with FLAG-Hsp90α WT alone or in combination with novobiocin (200 μM) to determine the effect of dimerization inhibition on Hsp90’s binding to Collagen-1 (Figure 7A). We observed that the presence of novobiocin reduced the binding of FLAG-Hsp90α WT to Collagen-1 compared with FLAG-Hsp90α WT alone, indicating that the inhibition of dimerization through the C-terminal binding of novobiocin reduces Hsp90’s binding capacity to Collagen-1.

To investigate the impact of Hsp90 dimerization inhibition on the alignment of Collagen-1 fibers, we treated Collagen-1 gels with a combination of Hsp90 and novobiocin at 200 μM (Figure 7B,C). We compared the alignment of the orientation of Collagen-1 fibers resulting from this combined treatment to that of untreated Collagen-1 gels. Our findings showed that Collagen-1 gels treated with FLAG-Hsp90α WT alone exhibited an expected increase in fiber alignment compared with NT gels. However, the combination treatment of FLAG-Hsp90α WT with novobiocin did not show a difference in fiber alignment compared with untreated gels. This suggests that the disruption of Hsp90 dimerization by novobiocin successfully inhibits Hsp90’s ability to align Collagen-1 fibers.

These observations indicate that the dimerization of Hsp90 may be a critical structural feature that enables Hsp90 to align and orient Collagen-1 fibers.

## 4. Discussion

Hsp90 is a protein of interest for cancer treatment; however, clinical trials have shown limited success due to the toxicity caused in healthy cells. Seeking alternative methods to inhibit Hsp90’s tumor-promoting role, aside from pan-Hsp90 inhibitors, offers a more practical means to minimize harm to normal cells.

eHsp90 was first discovered to be secreted by cancer cells by Eustace and colleagues, who found that Hsp90 secreted by cancer cells activated MMP2 and increased invasion [3]. The active MMP2 is implicated in cancer invasion by degrading the ECM surrounding the tumor cells [79,80,81]. Subsequent studies have not only substantiated the critical role of eHsp90 in enhancing invasion, but also discovered other extracellular interacting proteins that are proinvasive when in active state [3,5,8,10,24,25,27,82,83]. The eHsp90 was found to interact with extracellular proteins that that primarily are ECM proteins or ECM-modifying proteins, such as MMP-2, MMP-9, tPA, and LOXL2 [3,8,10,18,24,25,27,28,84]. Consequently, eHsp90 can be considered a pro-invasive hub that activates a cohort of proteins outside cancer cells to enhance invasion and metastasis, and its inhibition may be of benefit in treating metastatic cancers. Supportive of this idea is that several labs have used eHsp90 inhibitors in animal models of cancer metastasis and have shown benefits for reduced metastasis and increased survival [4,9,14,15,16,18,85,86]. This makes identifying new mechanisms by which eHsp90 makes the TME pro-invasive an important task to undertake to determine whether inhibiting eHsp90 may serve as a basis for anti-metastasis drugs. Other Hsps, such as Hsp40 and Hsp70, are also found in the extracellular environment, and they can enhance Hsp90’s interaction with MMP2, an enzyme involved in extracellular matrix remodeling [23]. Additionally, tumor-derived extracellular vesicles (T-EVs) expressing Hsp70 possess the capability to instigate anti-tumor immune responses [87]. As such, the Hsp70 or Hsp90-Hsp70 cohort may warrant future studies on their contribution to the TME.

eHsp90 was originally shown to be a critical regulator of the activation of ECM proteases, including MMP-2, MMP-9, and tPA, leading to increased invasion through an ECM composition that mimics the basement membrane, Matrigel [3,5,9,10,13,16,19,32,33]. This is not surprising given the importance of cancer cell interactions with their local ECM during invasion. However, the role of eHsp90 in invasion within the IM, which is predominantly composed of Collagen-1, has not been established. However, eHsp90 has also been demonstrated to interact with fibronectin, which, like Collagen-1, constitutes a component of the IM within the TME, albeit at a lower abundance than Collagen-1 [23,24,25]. The Hsp90–fibronectin interaction can prevent and influence matrix assembly and the endosomal internalization of fibronectin; however, whether it can lead to the invasion of cancer cells has not been explored [23,24,25].

In this study, we explored whether eHsp90 promotes the invasion in Collagen-1 as it does in Matrigel. We showed here that eHsp90 does indeed increase the invasiveness of breast cancer cells through the Collagen-1 matrix by increasing the alignment of Collagen-1 fibers making up the matrix. Aligned Collagen-1 fibers surrounding the tumors are a well-established marker of an invasive tumor [37,38,39,40,42,43,44,45].

Our results ruled out the idea that eHsp90 activates proLOXL2 and leads invasion and that Collagen-1 alignment relies on eHsp90–proLOXL2 interaction (our initial hypothesis). Instead, we found a novel direct interaction between Hsp90 and Collagen-1 that led to enhanced fiber alignment and invasion through Collagen-1-coated Transwells. Since Collagen-1 was not previously known to interact with eHsp90, we followed up on this observation by interrogating the mechanism by which eHsp90 functions to align the Collagen-1 fibers’ orientation. We showed that Hsp90 can bind directly to Collagen-1 through the N-domain. However, an isolated N-domain of Hsp90 is not sufficient to increase the invasion of breast cancer cells in the Collagen-1 matrix. Whether any of the Hsp90 domains are sufficient to increase the alignment of Collagen-1 needs to be tested via SHG microscopy by first treating the gels with GST-tagged Hsp90 N, M, and C domains. If required for further clarification of the mechanism, both Collagen-1 alignment and MB231 invasion can be tested after the cleavage of the GST tag from the purified Hsp90 domain proteins.

We also showed that, although the binding was led by the N-domain of Hsp90, neither the inhibition of ATPase activity nor the conformation of Hsp90 reliant on the N-domain impacted the binding of Hsp90 to Collagen-1. While inhibition of the ATPase activity of Hsp90 did not inhibit Hsp90-led Collagen-1 fiber alignment either, restricting Hsp90 to a closed conformation state did inhibit Collagen-1 alignment. A simple mechanism to explain Hsp90-led Collagen-1 alignment is that Hsp90 is a dimer, and the N-terminal domains each bind to a Collagen-1 fiber to bring the Collagen-1 fibers together. Consistent with this, the Hsp90 inhibitor novobiocin, which prevents dimerization, inhibited Hsp90-enhanced Collagen-1 alignment. However, confirming the effect of Hsp90 dimerization on Collagen-1 binding and alignment might require further validation using Hsp90 non-dimerizing constructs, such as the one described by Wayne et al. [74].

While Hsp90 by itself binds and aligns Collagen-1fibers, the extracellular factors present in the CM decreased the concentration of Hsp90 required for alignment by a hundred-fold. This suggests that Hsp90-induced Collagen-1 may be enhanced by other unidentified factors released by cancer cells. eHsp90 is known to interact with and activate an increasing number of pro-invasive proteins, and it will be interesting to identify such factors for its pro-invasive force in the interstitial matrix [20,80]

eHsp90 has been found to bind to fibronectin, and this interaction leads to increased matrix formation [23,24,25]. Interestingly, Chakraborty and colleagues did not observe any Hsp90-dependent changes in collagen morphology or assembly in a 2D context, whereas our study clearly demonstrated enhanced alignment of fibers in a 3D matrix using SHG imaging [25]. This disparity may be attributed to the higher sensitivity of SHG imaging in detecting structural changes and the greater accuracy of our 3D analytical methods in assessing Hsp90-induced fiber alignment [47]. Furthermore, unlike the immunofluorescence approach employed in their study, SHG imaging does not require ECM processing, which could induce fiber alignment regardless of Hsp90-α presence, potentially saturating the fiber assembly prior to the Hsp90 treatment of Collagen-1. Another factor to consider is that Chakraborty’s study used the β isoform of Hsp90 to show its ability to assemble fibronectin but not collagen fibers, whereas our findings demonstrate that the Hsp90-α isoform binds to Collagen-1 and enhances its alignment. The structural distinction between Hsp90’s α and β isoforms lies in the hinge region (the stretches between the N and M domains), suggesting that Hsp90-β might be unable to bind to Collagen-1 due to the N-domain’s involvement in this interaction.

Our current study uncovered a previously unknown role of eHsp90—that it facilitates the invasion of breast cancer cells within the Collagen-1 matrix by aligning Collagen-1 fibers. This novel function of eHsp90 implicates eHsp90, enabling the navigation of breast cancer cells even beyond the basement membrane within the Collagen-1-rich, longer-spanning matrix IM to invade surrounding tissues and reach the circulatory vessels, thus extending their reach beyond the initial tumor site. That inhibition of Hsp90 using 7191 decreases Collagen-1 binding and the alignment of Collagen-1 fibers, resulting in the invasion of breast cancer cells, suggesting that eHsp90 inhibition can be utilized therapeutically as an anti-invasion and metastasis approach. This is consistent with other studies using blocking antibodies or other small molecule inhibitors [8,16,18,86]. 7191 is a biotinylated form of the Hsp90 inhibitor Ganetespib, and because of the polarity of this moiety, it does not cross the cell membrane [28]. This easy chemical modification may be useful for other Hsp90 inhibitors that could be modified without affecting their binding or inhibitory activity.

While our study focused on the invasion of cancer cells, eHsp90-enhanced Collagen-1 alignment could also affect the migration of stromal cells, like immune cells, into the tumor periphery and enhance other metastatic steps, for example, creating a premetastatic niche (PMN) [88,89]. A PMN is a site on secondary organs that is modified in advance of cancer cell homing and colonization [89,90]. Serum and plasma levels of Hsp90 have been shown to be elevated in invasive cancers and are an established marker associated with poor outcomes for several cancers [8,8,91,92,93,94,95,96,97,98]. With elevated eHsp90, other organs could serve beyond the primary tumor microenvironment to align Collagen-1 around other tissues to enhance homing. Consistent with this, animal studies that have shown that blocking eHsp90 can reduce metastasis and improve survival, have been performed using a tail vein injection model for metastasis, however, tail vein injections do not allow the investigation of the initial steps of metastasis like invasion [7]. Therefore, inhibiting eHsp90 must reduce a metastatic step from surviving and exiting the circulation to homing and colonization. We speculate that eHsp90-led alignment of Collagen-1 in the interstitial matrix surrounding other organs may be part of this.

## 5. Conclusions

The data presented here provide conclusive evidence that Hsp90 plays a critical role in promoting the invasiveness of breast cancer cells through the Collagen-1 matrix by directly binding to and enhancing the alignment of Collagen-1 fibers. We ruled out the involvement of proLOXL2 activation in this process. Instead, we discovered that N- the domain of Hsp90 binds directly to Collagen-1, although its attributed activity of ATP hydrolysis is not crucial for binding and aligning Collagen-1 fibers. We showed that Hsp90’s open conformation is necessary for it to align Collagen-1 fibers and increase breast cancer invasion through Collagen-1. Although the isolated N-domain alone is not sufficient to increase invasion in the Collagen-1 matrix, our study suggests that Hsp90 dimerization, where likely each N-terminal domain binds to a Collagen-1 fiber, may be responsible for the observed alignment.

The implications of our findings are many. The They offer strategies for inhibiting extracellular Hsp90 (eHsp90) activity, which could effectively curb the invasion of breast cancer cells into connective tissues. Notably, these strategies are not limited to the primary tumor site but extend to secondary homing tissues, particularly those enriched with Collagen-1, such as metastatic and premetastatic sites. This research provides a promising avenue for the development of therapies aimed at intervening and inhibiting the progression of metastasis.

The study points to other aspects of investigation. For instance, Hsp90 dimerization and the role of individual domains in Collagen-1 alignment and identifying pro-invasive factors that enhance eHsp90-driven Collagen-1 alignment. These mechanistic studies will help design therapies targeted to inhibit eHsp90 led Collagen-1 invasion and metastatic progression.

## Figures and Tables

**Figure 1 cancers-15-05237-f001:**
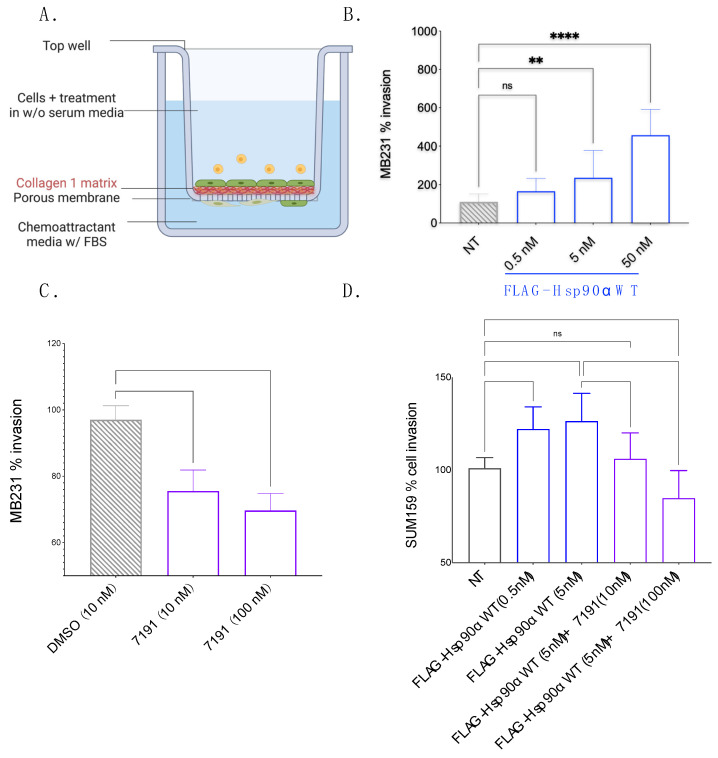
eHsp90 increases invasion of breast cancer cells through Collagen-1. (**A**). Schematic representation of Collagen-1-coated Transwell setup. The cells and treatment conditions mixed in the SS media were added to the top well. The cells invaded through the Collagen-1 matrix and the porous membrane toward the serum containing chemoattractant media. The cells that successfully invaded the other side of the membrane were counted. (**B**). MB231 invasion through Collagen-1 in a Transwell setup after cells in the top well were treated with exogenous FLAG-Hsp90α WT (one-way ANOVA *p* < 0.0001). (**C**). MB231 invasion through Collagen-1 in a Transwell setup after cells in the top well were treated with 7191 (one-way ANOVA *p*—0.0019) toward 2% FBS media for 8 h. (**D**). SUM159 invasion through Collagen-1 in a Transwell setup toward 1% FBS media for 4 h after treatment with FLAG-Hsp90α WT and 7191 (one-way ANOVA *p* < 0.0001). (**B**,**D**). *p*-value for pairwise comparison is depicted as—<0.1234 as “NS”, <0.0021 as “**”, and <0.0001 as “****”.

**Figure 2 cancers-15-05237-f002:**
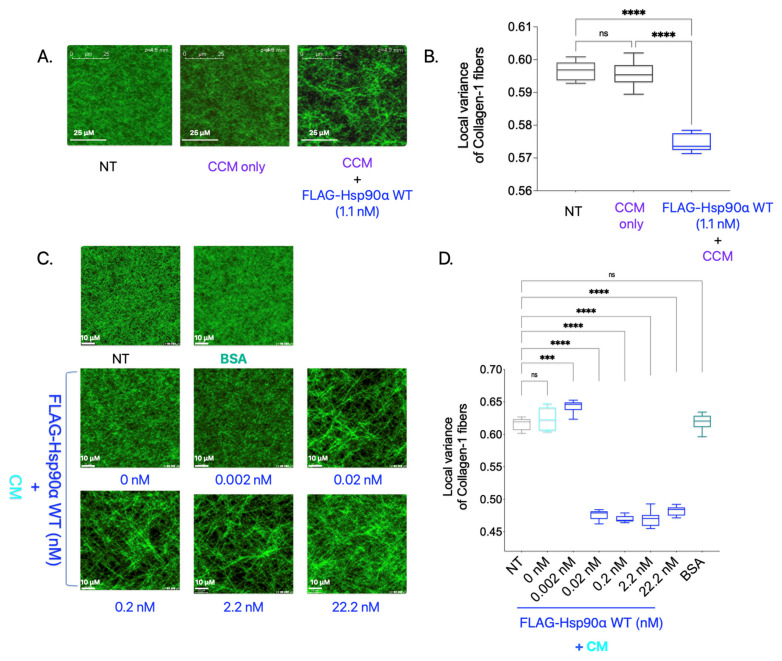
eHsp90 with extracellular factors can align Collagen-1 fibers. (**A**). SHG imaging experiment with purified protein treatment of polymerized Collagen gels along with 10× CCM mixed in PBS and incubated for 5 days at 37 °C. SHG images were acquired spanning a 70 μm vertical distance (1 image/0.5 μm) starting at 200 μm from the base of the gel. The images shown in the figure are representative images of average projections of z-stacks of each condition. The white line in the images is a scale bar depicting a 25 μm distance. (**B**). Quantification of local 3D variance of the Collagen-1 fibers measured from SHG-acquired images (one-way ANOVA; *p* < 0.0001). (**C**). Collagen-1 gels were treated with purified proteins along with MB231 CM mixed in PBS. The images were captured and z-stacked, as described in Figure 2A. The images shown are representative average intensity projections of z-stacks captured of each condition. White lines shown in the images are scale bars representing 10 μm. (**D**). Quantification of local 3D variance of the Collagen-1 fibers (one-way ANOVA; *p*-value < 0.0001). (**B**,**D**). *p*-value for pairwise comparison is depicted as—<0.1234 as “NS”, 0.0002 as “***”, and <0.0001 as “****”.

**Figure 3 cancers-15-05237-f003:**
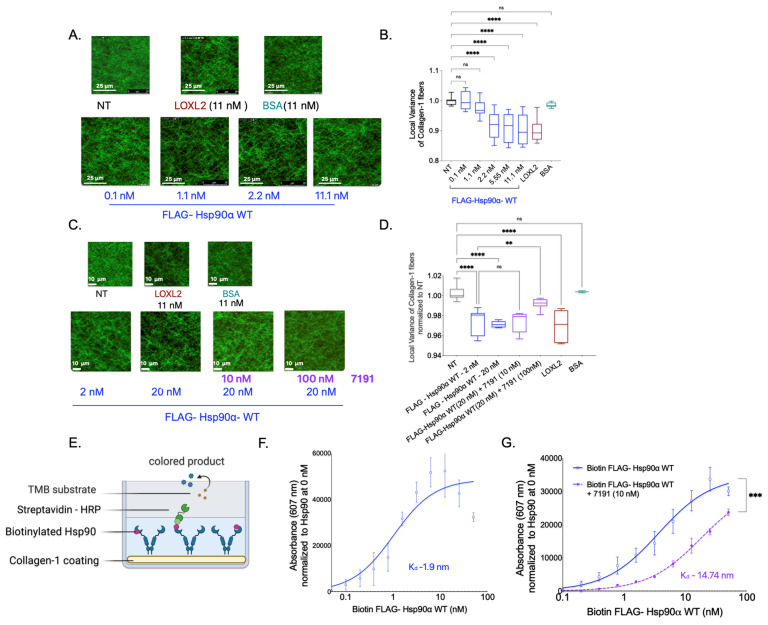
Hsp90 directly binds and is sufficient to align Collagen-1. (**A**). Collagen-1 gels were treated with purified FLAG- Hsp90α WT mixed in PBS and incubated for 5 days at 37 °C. LOXL2 was used as the positive control, and BSA was used as the negative control. SHG images were acquired spanning a 70 μm vertical distance (1 image/1 μm) starting at 200 μm from the base of the gel. The images shown in the figure are representative images of average projections of z-stacks of each condition. The white line in the images is a scale bar depicting a 25 μm distance. (**B**). Quantification of local variance of the Collagen-1 fibers analyzed from z-stack images obtained (one-way ANOVA; *p*-value < 0.0001). (**C**). Collagen-1 gels were treated with FLAG- Hsp90α WT and 7191 mixed in PBS and incubated for 5 days at 370. Controls were used as in Figure 3A. SHG images were acquired on day 5, spanning 70 μm with 1 image/1 μm. The figure shows representative average intensity projections of z-stacks of images captured for each condition. White lines shown in the images are scale bars representing 10 μm. (**D**). Quantification of local variance of the Collagen-1 fibers in z-stack images acquired (one-way ANOVA; *p*-value < 0.0001). (**B**,**D**). *p* value for pairwise comparison is represented as follows—<0.1234 as “NS”, <0.0021 as “**”, and <0.0001 as “****”. (**E**). Schematic of Collagen-1 binding assay. (**F**). A collagen-1 binding assay with recombinant FLAG-Hsp90α WT was performed. Linear regression of absorbance values with one site-specific binding is plotted, Kd = 1.9 nM. (**G**). Collagen-1 binding assay with FLAG-Hsp90α WT alone and FLAG-Hsp90α WT incubated with 10 nM of 7191. Linear regression with one site-specific binding—Kd = 14.74 nM (paired *t*-test; *p*-value = 0.0006). (**F**,**G**). *p* value for pairwise comparison is represented as follows—0.0002 as “***”.

**Figure 4 cancers-15-05237-f004:**
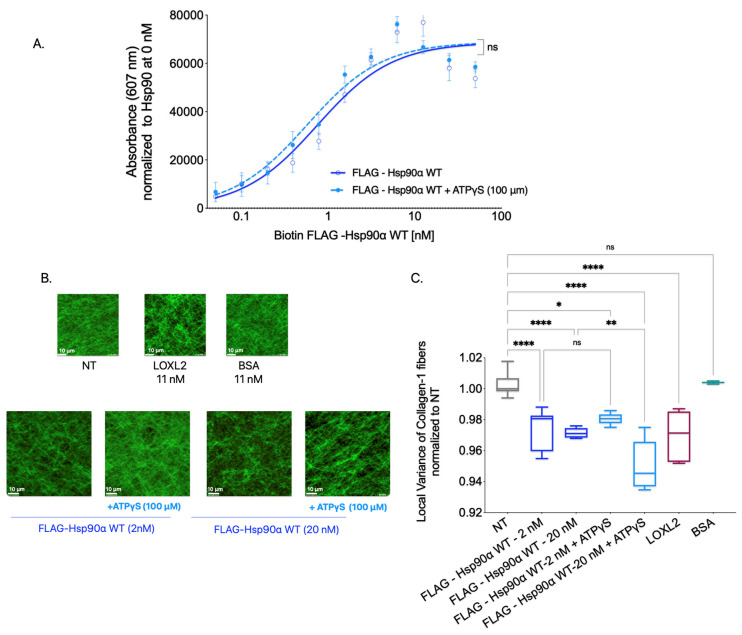
Hsp90’s ATPase activity is not critical for binding and aligning Collagen-1; however, slowing ATPase activity increases Collagen-1 alignment. (**A**). Collagen-1 binding assay with FLAG-Hsp90α WT alone and combined with 100 μM ATPγS. Absorbance was measured at 607 nM and normalized to absorbance at FLAG- Hsp90α WT treatment at 0 nM. Linear regression of absorbance values with one site-specific binding is plotted (paired *t*-test; *p*-value < 0.0001). *p* value for pairwise comparison is represented as follows—<0.1234 as “NS”. (**B**). Collagen-1 gels were treated with FLAG- Hsp90α WT and 100 μM ATPγS mixed in PBS and incubated for 5 days at 37 °C. LOXL2 (positive) and BSA (negative) treatments were controls. SHG images were acquired on day 5, spanning 70 μm (1 image/1 μm) starting at 200 μm above the base of the gel. The images shown in the figure are representative images of average projections of z-stacks of each condition. The white line in the images depicts a 10 μm distance. (**C**). Quantification of local fiber variance of Collagen-1 gel images (one-way ANOVA; *p* < 0.0001). *p* value for pairwise comparison is represented as follows—<0.1234 as “NS”, <0.0332 as “*”, <0.0021 as “**”, and <0.0001 as “****”.

**Figure 5 cancers-15-05237-f005:**
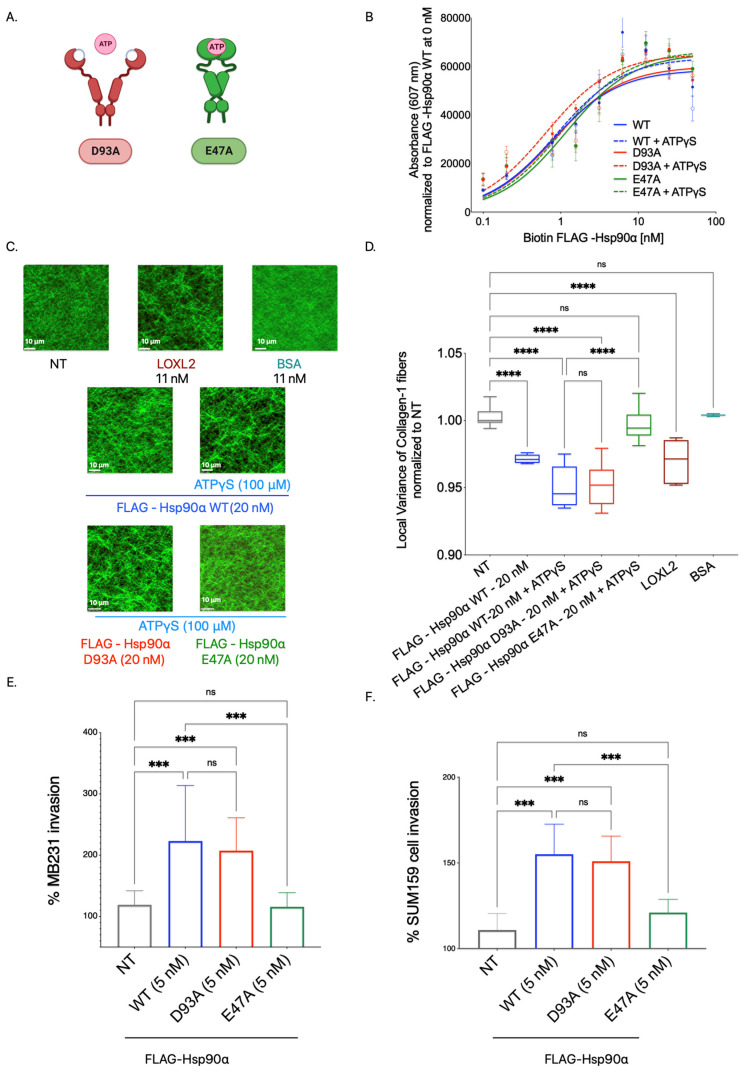
Hsp90’s binding to Collagen-1 is independent of its conformation, but open conformation is essential for aligning Collagen-1 fibers and breast cancer cell invasion through Collagen-1. (**A**). Schematic of Hsp90’s conformation mutants. (**B**). Collagen-1 binding assay with biotinylated FLAG-Hsp90α WT, D93A, and E47A with and without ATPγS (100 μM). Absorbance measured at 607 nM is normalized to the absorbance of FLAG-Hsp90α WT at 0 nM. One-way ANOVA *p* < 0.0001. (**C**). The conditions were prepared in PBS with and without ATPγS (100 μM) and added to wells containing polymerized Collagen-1 gels. SHG images were acquired spanning a vertical distance of 70 μm (starting at 200 μm above the base of the gel) with 1 image/1 μm. The images shown are representative images of the average intensity projection of z-stacks of images acquired via SHG. The white line in the images depicts 10 μm. (**D**). Quantification of local variance in Collagen-1 fiber alignment (one-way ANOVA with Tukey’s correction for multiple comparisons, *p* < 0.0001). *p* value for pairwise comparison is represented as follows—*p* < 0.1234 as “NS”, *p* < 0.0001 as “****”. (**E**,**F**). Collagen-1 Transwell invasion assay with MB231 cells and SUM159 cells. Invaded cells were imaged and counted. *p* value for pairwise comparison is represented as follows—*p* < 0.1234 as “NS”, *p* < 0.0002 as “***”. (**E**). Collagen-1 Transwell invasion assay with MB231 (one-way ANOVA; *p* = 0.0088). (**F**). Collagen-1 Transwell invasion assay with SUM159 cells (one-way ANOVA; *p* = 0.001).

**Figure 6 cancers-15-05237-f006:**
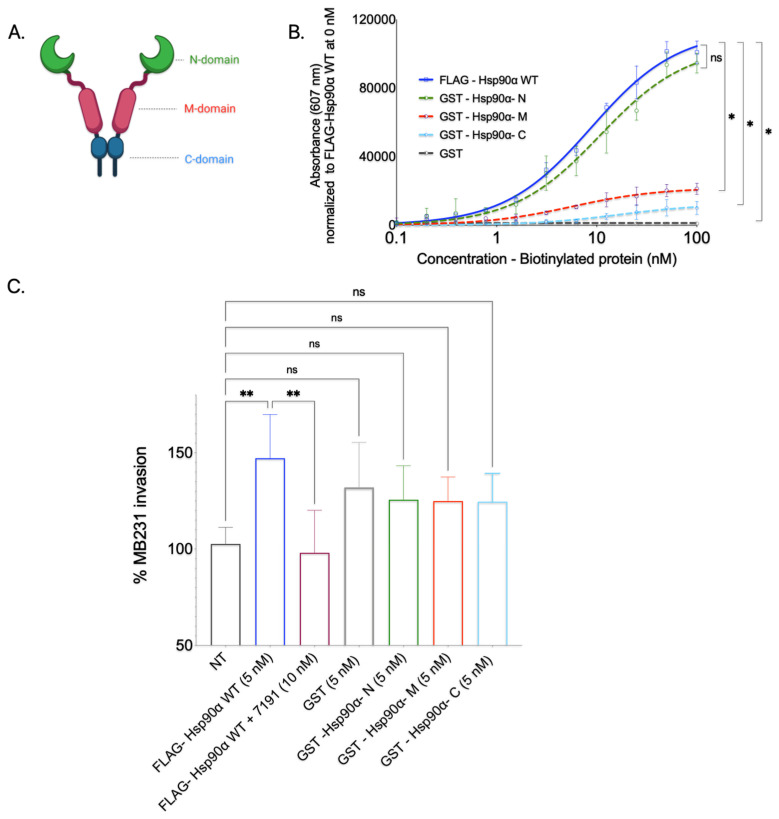
Hsp90 binds to Collagen-1 via the N-domain, but separate domains are insufficient to increase cancer cell invasion through Collagen-1. (**A**). Schematic of Hsp90 structure and its domains. (**B**). Collagen-1 binding assay with purified GST-tagged Hsp90 domain proteins. Absorbance measured at 607 nM is normalized to the absorbance of FLAG-Hsp90α WT at 0 nM. One-way ANOVA *p* = 0.0077. *p* value for pairwise comparison is represented as follows—*p* < 0.1234 as “NS”, *p* < 0.0332 as “*”. (**C**). Collagen-1 Transwell invasion assay with MB231 cells treated exogenously with GST tagged N, M, and C domains of Hsp90. One-way ANOVA was performed with Tukey’s correction; *p*-value—0.0010. *p* value for pairwise comparison is represented as follows—*p* < 0.1234 as “NS”, *p* < 0.0021 as “**”.

**Figure 7 cancers-15-05237-f007:**
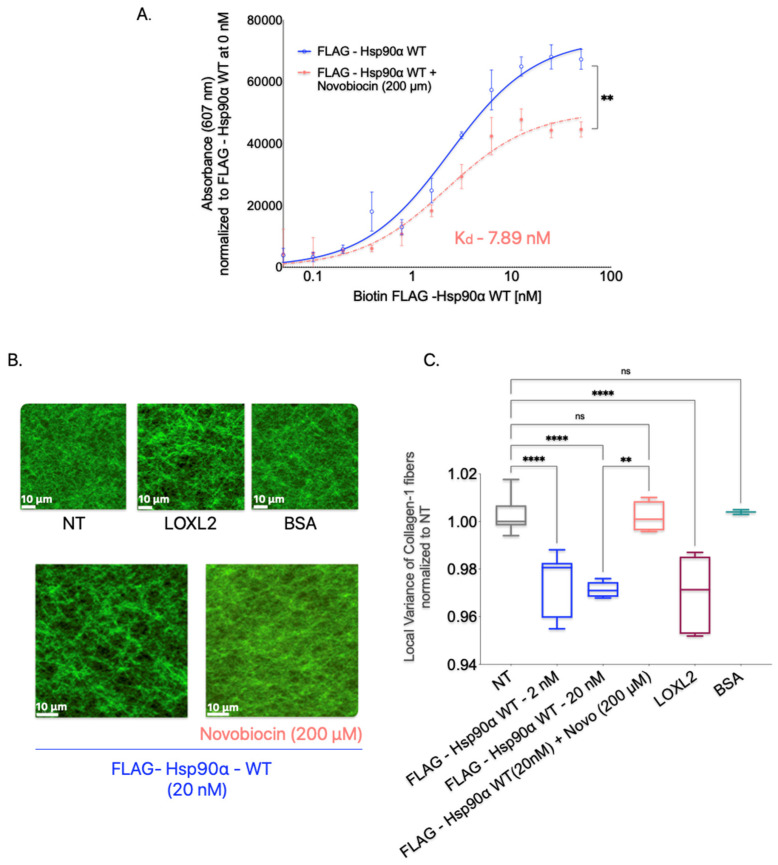
C-terminal dimerization is critical for Hsp90’s binding to and aligning Collagen-1 fibers. (**A**). Collagen-1 binding assay with purified Hsp90 treated with novobiocin. Absorbance measured at 607 nM is normalized to the absorbance of FLAG-Hsp90α WT at 0 nM. One-way ANOVA *p* = 0.0049. *p* value for pairwise comparison is represented as follows—*p* < 0.0021 as “**”. (**B**). The conditions were prepared in PBS and added to the wells containing polymerized Collagen-1 gels and incubated for 5 days at 37 °C. SHG images were acquired spanning a vertical distance of 70 μm with 1 image/1 μm. The images shown are representative images of the average intensity projection of z-stacks of images acquired via SHG. The white line depicts 10 μm. (**C**). Quantitation of local variance in Collagen-1 gel images acquired via SHG. Novo is Novobiocin. One-way ANOVA *p* < 0.0001. *p* value for pairwise comparison is represented as follows—*p* < 0.1234 as “NS”, *p* < 0.0021 as “**”, *p* < 0.0001 as “****”.

## Data Availability

Not applicable.

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
