# Peer review of "Extracellular Hsp90 Binds to and Aligns Collagen-1 to Enhance Breast Cancer Cell Invasiveness"

_cancers, 2023, doi:10.3390/cancers15215237_

Round 1
Reviewer 1 Report
Very well constructed set of experiments.
1. Have the authors looked at any other HSPs that could affect tumor microenvironment?
2. Since collagen1 is not the only protein making up ECM, how does eHSP90 affect other ECM proteins?
3. Since cancer cells produce high amounts of HSP90, does secretion of HSP90 into the ECM constitute majority of its mechanism for increase of invasiveness? How much is known about HSP90 affecting cancer cell motility dynamics? Do we know if high intracellular HSP90 increases cell motility per se?
4. It would be good to see if these findings can be replicated in an in vivo model.
Good work!
Author Response
- Have the authors looked at any other HSPs that could affect the tumor microenvironment?
Hsps found in the tumor microenvironment?
In the current investigation, although we did not specifically explore other Hsps present in conditioned media that enhanced the Hsp90-mediated Collagen-1 alignment, Hsp90 co-chaperones, including Hsp70 and Hsp40, may be influential contributors to this phenomenon. We will add the following to the manuscript.
Line 811- 816: “Other Hsps, such as Hsp40 and Hsp70, are also found in the extracellular environment, and they can enhance Hsp90’s interaction with MMP2, an enzyme involved in extracellular matrix remodeling [23]. Additionally, Tumor-derived Extracellular Vesicles (T-EVs) expressing Hsp70 possess the capability to instigate anti-tumor immune responses [1]. As such, the Hsp70 or Hsp90-Hsp70 cohort may bear future study in their contribution to the TME.”
- Since collagen1 is not the only protein making up ECM, how does eHSP90 affect other ECM proteins?
eHsp90 interacts with Fibronectin present in the interstitial matrix as well. The following statements have been added to the discussion.
Line 822-827: “eHsp90 has also been demonstrated to interact with Fibronectin, which, like Collagen-1, constitutes a component of the IM within the TME, albeit in lesser abundance compared to Collagen-1.” The Hsp90-Fibronectin interaction can prevent and influence matrix assembly and endosomal internalization of Fibronectin; however, it has not been explored whether it can lead to invasion of cancer cells. [2]–[4]
- Since cancer cells produce high amounts of HSP90, does secretion of HSP90 into the ECM constitute the majority of its mechanism for the increase of invasiveness?
While Intracellular HSP90 is increased markedly in cancer cells, thus far, its role in cancer migration has been shown to be indirect via its client proteins [5] [6]. Also, clinical trials for intracellular Hsp90 have failed, perhaps because of its roles in all cells.
- It would be good to see if these findings can be replicated in an in vivo model.
While we acknowledge that in-vivo studies are valuable for further research, In-vivo studies were beyond the scope of the current study that focused on mechanistic insights.
Reviewer 2 Report
The manuscript "Extracellular Hsp90 binds to and aligns Collagen-1 to enhance breast cancer cell invasiveness" by Singh et. al. studies the role of eHsp90 towards increasing the aggressiveness of breast cancer cell lines. The study is excellent from an in vitro point of view since the entire work was done in cultured cells. The authors have meticulously planned and executed the experiments and the data is quite sound. I have a concern below:
1. If possible, can the authors show in nude mouse model, the association of eHsp90 with collagen-1 fibers in the aggressive tumors formed by injecting 231 cells?
I would recommend for a major revision. Thank you.
Author Response
The manuscript "Extracellular Hsp90 binds to and aligns Collagen-1 to enhance breast cancer cell invasiveness" by Singh et. al. studies the role of eHsp90 in increasing the aggressiveness of breast cancer cell lines. The study is excellent from an in vitro point of view since the entire work was done in cultured cells. The authors have meticulously planned and executed the experiments, and the data is quite sound. I have a concern below:
- If possible, can the authors show the association of eHsp90 with collagen-1 fibers in the aggressive tumors formed by injecting 231 cells in a nude mouse model? I would recommend a major revision. Thank you.
In-vivo studies were beyond the scope of the current study that focused on mechanistic insights.